



**Technical note: Efficient imaging of hydrological units below lakes and fjords with a floating, transient**
**electromagnetic system (FloaTEM)**
Pradip Kumar Maurya[1], Frederik Ersted Christensen[1], M. Andy Kass[1], Jesper B. Pedersen[1], Rasmus R. Frederiksen[1],
Nikolaj Foged[1], Anders Vest Christiansen[1], and Esben Auken[1,2]
[1]Department of Geoscience, HydroGeophysics Group Aarhus University, C.F. Møllers Alle, 4, Aarhus C, Denmark
[2]The Geological Survey of Denmark and Greenland (GEUS) Oester Voldgade 10 1350 Copenhagen K Denmark, formerly at 1.
Corresponds to: pradip.maurya@geo.au.dk
**Abstract**
Imagining geological layers beneath lakes, rivers, and shallow seawater provides detailed information critical for
hydrological modelling, geologic studies, contaminant mapping, and more.  However, significant engineering and
interpretation challenges have limited the applications, preventing widespread adoption in aquatic environments. We have
developed a towed transient electromagnetic (tTEM) system to a new, easily configurable floating, transient
electromagnetic instrument (FloaTEM) capable of imaging the subsurface beneath both fresh and saltwater water bodies.
Based on the terrestrial tTEM instrument, the FloaTEM system utilizes a similar philosophy of a lightweight towed
transmitter with a trailing, offset receiver, pulled by a small boat. The FloaTEM system is tailored to the specific fresh or
saltwater application as necessary, allowing investigations down to 100 m in freshwater environments, and up to 20 m on
saline waters. Through synthetic analysis we show how the depth of investigation of the FloaTEM system greatly depends
on the resistivity and thickness of the water column. The system has been successfully deployed in Denmark for a variety
of hydrologic investigations, improving the ability to understand and model processes beneath water bodies. We present
two freshwater applications and a saltwater application. Imaging results reveal significant heterogeneities in the sediment
types below the freshwater lakes. The saline water example demonstrates that the system is capable to identify and
distinguish clay and sand layers below the saline water column.
**1. Introduction**
Understanding interactions between surface water and groundwater is necessary for effective management of water
resources as they are both part of an interconnected hydrologic system (Sophocleous, 2002; Winter et al., 1998; Harvey
and Gooseff, 2015).  This requires knowledge of hydrogeological settings below the water column of lakes, streams, and
other water bodies, in addition to properties underlying adjacent onshore areas. Non-invasive geophysical methods
provide spatial information on these subsurface properties and processes across many environments; over the last few
decades the methods have played a vital role in near-surface investigations (Barker, 1980; Hatch et al., 2010a; Day-Lewis
et al., 2006).  However, deployment of surface-based geophysical investigations (as opposed to airborne systems) on
water bodies has historically been difficult (Sheets and Dumouchelle, 2009; Briggs et al., 2019; Parsekian et al., 2015);
while not insurmountable, this has limited the application range to some degree.
Electrical and electromagnetic methods (EM) are the two most-extensively used geophysical exploration and
characterization techniques for hydrologic applications (Binley and Kemna, 2005; Danielsen et al., 2003; Christiansen et





al., 2006; Auken et al., 2003; Minsley et al., 2021; Siemon et al., 2009). While classically used on land, several studies have shown that these methods can also be used on lakes, streams, or rivers. Among the electrical methods, electrical resistivity tomography (ERT) has been a common and robust technique, with applications to aquatic environments including mapping the distribution of clay sediments, mapping freshwater saturation in saltwater bay sediments (Manheim et al., 2004), and estimating sediment thicknesses and locating faults (Kwon et al., 2005). These studies deployed relatively long floating cable layouts, or streamers, of approximately 100 meters, towed by a boat for collecting continuous resistivity data. Longer cable layouts, giving deeper information, limit the operational efficiency significantly. This implies that these instruments inherently have a limited depth of investigation (DOI).

Applications of transient electromagnetic (TEM) and frequency domain EM tools are reported in previous studies, e.g., discharge of groundwater to lakes and brines (Ong et al., 2010; Briggs et al., 2019), and extraction of lithium from large scale natural brine systems (Munk et al., 2016). Airborne techniques have proved capable of mapping beneath lakes, rivers and near-shore seas (Fitterman and Deszcz-Pan, 1998; Dickey, 2018; Rey et al., 2019), but are costly and provide lower vertical and lateral resolution than their ground-based counterparts (Hatch et al., 2010b).

There has been a growing interest in the development of a towed, waterborne EM system, as such an instrument provides continuous information with high lateral resolution. Mollidor et al. (2013) have shown an application of a commercial in-loop transient EM (TEM) system on a volcanic lake to map sediment thickness. Since the system had a large transmitter loop (18x18 $m^2$), they encountered non 1D-effects requiring 3D modelling for proper interpretation. Hatch et al. (2010b) presented results from a waterborne survey where they used a floating setup of a commercial TEM system, used over a 40 km section of the Murray River (Australia) to monitor the influx of saline water. These studies and systems, while effective, have limitations preventing their widespread use in waterborne applications, specifically in terms of limited DOI and horizontal resolution. An ideal system would be compact and lightweight, have a small footprint, and provide sufficient transmitter power to investigate the hydrogeological properties beneath the water column.

Recent advancements in electronics of EM instrumentation led Auken et al. (2018) to develop a ground-based towed transient electromagnetic system (tTEM) for efficient and high resolution 3D mapping of the subsurface (Maurya et al., 2020). The tTEM system provides the necessary framework for creating a floating, towed EM system. The tTEM-system is relatively compact, with the entire system extending no more than 16 m behind the towing vehicle and a maximum width of 4 m. It has high lateral resolution, down to 10 m x 10 m. The tTEM also has a relative high transmitter moment for such a compact system, providing depths of investigation in ground-based surveys down to 100m. The waterborne version of the tTEM system is referred to as FloaTEM (see Fig.1) and a recent application of the FloaTEM system has been presented by Lane et al. (2020) where they successfully used the ground configuration of the system on rivers and estuaries in the United States to characterize the underlying hydrological system. In their study the system was used as it was designed for ground-based applications (Auken et al., 2018) without any modifications to actual geometry and measurements protocols. In this paper, we present a greatly improved and a flexible version of the FloaTEM system to investigate subsurface properties beneath both fresh and saline water columns. We highlight the design aspects of the system and discuss capabilities and limitations. Finally, we present three case studies to demonstrate the efficacy of the FloaTEM system and interpretation methodology: surveying on a shallow freshwater lake, a deep freshwater lake, and in a saline bay environment.



## 2. The FloaTEM system

Operating in aquatic environments provides challenges that are unique to the setting, requiring modifications not only to the instrumentation relative to land-based operation, but also to acquisition protocols and safety procedures. Navigating on shallow water, lakes, or rivers, may be challenging; to assist safe navigation, real time GPS and echo-sounder data are integrated into the FloaTEM system's recording and navigation software. The echo-sounder provides the depth to the river/lakebed and this information can furthermore be utilized as prior information in later data processing.

Design aspects of the FloaTEM system depends on the application—primarily whether freshwater or saltwater—and thus we have designed both a fresh water FloaTEM system (FW-FloaTEM) and a saltwater FloaTEM system (SW-FloaTEM). In the following, we discuss the details of freshwater and saltwater FloaTEM systems.

### 2.1 The freshwater FloaTEM system

The FW-FloaTEM has a design similar to the tTEM-system: A 4 x 2 $m^2$, single-turn transmitter coil (TX-coil) is followed by the receiver coil (RX-coil), in a 9 m offset configuration. Figure 1 shows a schematic layout and photo of the FW-FloaTEM system. The receiver coil has an effective area of 20 $m^2$ with a bandwidth of 420 kHz. This effective RX-area is 4 times higher compared to the previously used RX-coil of the tTEM-system as described in Auken et al. (2018), and therefore provides approximately a 4 times better signal to noise ratio and increased DOI (100m).

The fiberglass frame follows the same construction as the tTEM-system—mounted on two paddleboards instead of sleds—and with additional frame components added for stability. The RX-coil is simply mounted on an inflatable rubber boat. Note that all mounting and floatation devices of the TX- and RX-coils are of non-conductive materials to avoid EM bias signals in the data.

The acquisition protocol consists of an alternating high- and low-moment transmitter pulse to obtain the sounding curve. The low moment, with a peak current of ~3 A, records 15 time gates of data between 4 µs and 33 µs referenced to the beginning of the turn-off of the transmitter pulse. The high-moment pulse utilizes 23 gates from 10 µs to 900 µs with a peak current of ~30 A. Thanks to the latest hardware modification, the peak current is maintained with a deviation of ±0.1A, which ensures a stable current waveform throughout the operation. Detailed system parameters are listed in Table-1.

### 2.2 The saltwater FloaTEM system

Presence of highly conductive saltwater limits the DOI due to the slow diffusion of the eddy currents in the conductive water body. In order to increase the DOI, the transmitter moment of the SW-FloaTEM is increased by a factor of eight, compared to FW-FloaTEM, by doubling the transmitter loop size and increasing the number of TX-coil turns to four. The saltwater configuration only utilizes a HM pulse of 25 A which is sufficient to obtain similar near surface resolution as the freshwater system since the long-duration eddy currents in conductive seawater obviate the need to record very early times. Further justification for using only HM is given in the synthetic studies section. Table-1 shows the parameters for FW- and SW-FloaTEM systems. Observe that the last measurement gate for SW-FloaTEM is ~3 ms compared to ~1 ms for FW-FloaTEM system.

The signal to noise ratio (S/N) is further increased by using a 40m2 RX coil. As the limiting factor for these RX coils is the noise in the pre-amplifier (Nyboe and Sørensen, 2012) increasing the area of the coil increases the S/N ratio





proportionally. This is true as long as the area is below approximately 200 m2. Hence, the total S/N ratio increase for the
SW-FloaTEM system compared to the FW-FloaTEM system is a factor of 8 for the peak moment and a factor of two for
the RX- coil, in total a factor of 16.

**3. Model resolution study**
A model resolution study was conducted to investigate the influence of water depth and water conductivity on the
resolving capabilities of FloaTEM systems for the sub-water layers. The focus of the resolution study was the case of a
saltwater environment, where the conductive water layer limits the DOI significantly, and decreases the resolution of sub-
water resistivity structures. The model resolution study was also used in the design of the SW-FloaTEM system, and the
presented results therefore include both the FW- and SW-FloaTEM cases. The model resolution study comprises a) an
inversion of synthetically generated data from known layered models (the *true model*); b) a model parameter analysis of
the true models, and c) an estimated depth of investigation (DOI). The modelling was performed with a 1D framework,
and hence does not examine lateral resolution capabilities or ability to resolve 2D or 3D structures.
The modelling scheme consists of the following steps:

123        1.  Calculate system-specific 1D forward data of the true model.

124        2.  Estimate realistic data uncertainties on the forward data based on signal levels and background noise assumptions

125        3.  Estimating model parameter uncertainties by a computation of the model covariance matrix for the true model.

126            (Auken et al., 2015)

127        4.   Performing 1D smooth inversions of the forward data including DOI estimates.

All the modellings were carried out with the AarhusInv modelling code (Auken et al., 2015). The FW- and SW-FloaTEM
systems were modelled as described in Table 1. The data uncertainty was model dependent, based on a background noise
level at 1nV/m$^2$ at 1 ms plus a uniform contribution of 3%. The uniform uncertainty is the main contribution to data
uncertainty due to the relatively conductive models producing high signals. For the model parameter analysis, a priori
constraints on the water column were applied with a 10% uncertainty for the water depth and a 30% uncertainty for the
resistivity of the water. For the inversion, no lateral constraints were applied. However, for the model parameter analysis
lateral constraints were assumed between 5 similar neighboring models (based on the true model) to simulate the improved
resolution capabilities from information sharing when working with field data. For the inversion of the forward data, a
smooth 30-layer model description was used with logarithmic increasing layer thicknesses with depth, and with an
additional top layer representing the depth and resistivity of the water column. All inversions were carried out using a
homogenous starting resistivity model.
Two model sweeps were constructed, each consisting of 15 three-layer models (True models). In model sweep 1 (Fig.
2a), the thickness (water depth) was varied of a 0.3 Ωm top layer from one to 15 m. In model sweep 2 (Fig. 3a) the
resistivity of a 7 m thick water layer was varied from 0.1 - 3 Ωm. In both model sweeps, the second layer was 3 Ωm / 10
m thick, and the third layer 30 Ωm.





The modeling results for model sweep 1 are shown in Fig. 2. Since the modeling was carried out in log-model space, the
model parameter analysis (Fig. 2b and 2c) shows the relative uncertainties estimates (STD-factor) of the model
parameters. In general, a model parameter (resistivity or thickness) will be considered resolved if the STD-factor is less
than 1.5, moderately resolved if between 1.5-2.0 and unresolved if greater than 2. From the model parameter analysis in
Fig. 2b and 2c, we observe, as expected, that the resolution of the model in general decreases with increasing water depth.
The water layer is very well resolved in all cases partly because of the prior constraints and partly due to the method's
high sensitivity to the conductive water layer. In the SW-FloaTEM case (Fig. 2b) the resistivity of the second layer is
resolved (STD-factor < 2) to a water depth of about 7 m and the layer boundary between layer two and three (DEP 2) is
resolved to a water depth of about 10 m. In the FW-FloaTEM case the (Fig. 2c) the resistivity of the second layer is
resolved (STD-factor < 2) to a water depth of about 5 m and the layer boundary to a water depth of only around 4 m.
Also, for the third layer, the resistivity was better resolved in the SW-FloaTEM system case than in the FW case.
The inversion results of the true model data, with DOI estimates in Fig. 2d and 2e, are in-line with the observations from
the model parameter analysis. Increasing water depth results in a shallower DOI and loss of resolution of the sub-water
layers, and the SW-FloaTEM system performs better than the FW-FloaTEM system.
Water depth is not the only parameter of importance for the resolution capabilities, but also the resistivity or conductivity
of the water. Figure 3 shows the modeling results for model sweep 2 with a varying resistivity of a 7 m thick water layer.
For a very conductive water layer of 0.1-0.2 $\Omega$m, the resolution is limited for both systems, as observed in the model
parameter analysis as well as in the inversion sections of Fig. 3. When the water resistivity is above 0.3-0.4 $\Omega$m, the SW
system resolves/recovers the sub-water layers very well (Fig. 3b and 3d). Especially in resolving the boundary between
second and third layer (DEP2), the SW-system performs much better than the FW-system, which is also clearly reflected
in the DOI of the two systems.
Based on the presented analysis and other analyses (not shown in this paper), we conclude that the conductance (product
of conductivity and thickness) of the water column should be below approximately 25 Siemens for this particular SW-
FloaTEM system to able to penetrate the water column and map sub-water layers. It was also clear that the S/N ratio for
the SW system had to be increased significantly compared to the FW system, but the very early time gates were not
needed, and a slower turn-off and lower bandwidth of the RX-coil was acceptable. This led to the compromise of more
turns in the transmitter coil, only high moment cycles and the larger area of the RX-coil.
**4. Field cases**
We present three surveys conducted with the FloaTEM system in Denmark: Two on freshwater lakes, and one on seawater
in a fjord. These datasets represent different water conductivities and various glacial sediment settings.  Details of
processing and inversion of FloaTEM data are given in appendix-A. Some of the cases needed special handling of the
inversion process and this is described in the respective case study section. Table 2 summarizes key survey conditions
and modeling parameters.
**4.1 Freshwater cases**
We present two freshwater cases from two lakes in central Jutland, Denmark, to demonstrate the utility of the FloaTEM-
FW system in a shallow and a deep lake scenario.



### 4.1.1 Lake Sunds

Lake Sunds spans 127 hectares and is quite shallow (1.5 m - 2.5 m) with a maximum depth of 4.5 m. It is sitting in a late Weichselian meltwater plain. The City of Sunds has developed around the lake, and the majority of the ~4000 inhabitants of Sunds live close to the water. In recent years the groundwater table in Sunds has risen substantially, which causes problems in the winter period where the groundwater is the highest and periods of heavy rain then results in flooded cellars in residential houses. The problem is exacerbated by an old sewage system in the city with many worn pipes. These pipes are under replacement, but this will remove the current drainage by worn pipes, and the consequences would be a further rise of the groundwater table. On top of the flooding of cellars, there is a risk that the groundwater fluctuations can mobilize near-surface pollutants from otherwise hydrologically inactive point-source pollutions in the city such as old gas stations and landfills and hence contaminate the groundwater in the area.

From a hydrogeological viewpoint the shallow water table has puzzled the water managers as shallow boreholes from the area show that the geology in the upper 20 meters is pure sand as expected in a meltwater plain environment. It was therefore decided to setup a detailed groundwater model to investigate groundwater flow paths and identify measures to control the groundwater table fluctuations.

The area to the east of the lake has been mapped with tTEM, covering a total of 816 hectares, with a FloaTEM survey subsequently performed on the lake (Fig. 4). Additionally, multiple boreholes provide lithological data for comparison, although the majority of the boreholes only reach 10-20 meters depth. Most of them were drilled in the 1940's in connection to brown-coal mapping.

The tTEM and FloaTEM data were inverted separately, with the results combined in Figure 5. Profile A in Fig. 4 is entirely on the lake and profile B is oriented north-south crossing the lake. In profile A, FloaTEM inversion results generally show a good agreement with the available borehole description (B1 and B2) which is broadly categorized as sand, clay and silt containing organic material. However, there is a slight mismatch between lithological boundaries observed in some boreholes and inversion models. This mismatch may be caused by borehole offset from FloaTEM profiles, possibly exaggerated by erroneous location data for the more than 70 years old logs. The distance of Borehole B1 and B2 from the profile are 20 and 25m. Overall, the resistivity model indicates a presence of two areas with a thick organic silt layer below the water column (Fig. 5a and 5c) followed by a thick and more resistive sand layer. The sand layer thins out towards the bank of the lake and appears to go to the surface outside the lake as indicated in profile B. The information about thickness and location of the organic silts are of great importance in the groundwater model of the area, since these old lake deposits are impermeable and thereby guide groundwater flow beneath the lake.

Figure 5c-f shows mean resistivity maps at four depth intervals and includes both the FloaTEM and the tTEM survey results. The mean resistivity maps indicate that there is a large degree of spatial variability of sediment types in and around Lake Sunds. The heterogeneity beneath the lake would not be possible to resolve by interpolating across; this heterogeneity is related to the lake genesis and reveals where the water table beneath the town of Sunds is in hydrologic contact with the lake. Furthermore, the tTEM and FloaTEM results show that the geological setting is not a simple sandbox at depth. At 20 meters depth and below we have several Tertiary clay layers with a resistivity of 10-30 ohm-m, which have been deformed by glaciers and glacial tectonics. The information about the clay layers is crucial for the deeper parts of the groundwater model.

### 4.1.2 Lake Ravn


Lake Ravn is located in Eastern Jutland, Denmark. It is the second deepest lake in Denmark with depths generally ranging
from 25 to 30 m, and with a maximum depth of 34 m. The lake was formed as a dead-ice hole located on top of a WSW-
ENE oriented partly-buried valley (Sandersen, 2016).
In the rOpen project (https://hgg.au.dk/projects/ropen), the Javngyde watershed northwest of Lake Ravn was mapped in
detail with tTEM, and it was modelled with a 3D finite difference groundwater flow model. The purpose of the rOpen
project was to estimate the total amount of nitrate reduction along flow pathways from the water table to a surface water
recipient. The rOpen work and a related hydrological modelling study (Rumph Frederiksen and Molina-Navarro, 2021)
revealed that around 40% of the infiltrating water crossed the surface watershed as groundwater flow to Lake Ravn.
However, the hydraulic connectivity between the watershed and the lake was poorly understood, and it was decided to
perform a FloaTEM survey on the lake to obtain more information about the hydrological system.
The survey was conducted with east/west oriented lines with a spacing of 60 m combined with lines encompassing the
perimeter of the lake (Fig. 6). Only electric boat engines are allowed on the lake, limiting the acquisition speed to 6 km/h.
Strong winds on the day of acquisition further challenged the navigation and resulting in head-wind lines being wigglier
than the tail-wind lines.
The resistivity model for Lake Ravn (Figure 7) shows multiple features of interest. The relatively high resistivity of the
lake water has allowed for extended depths of investigation, despite the deep-water column. The resistivity models have
a DOI down to 90 m below the lake surface. Within the water column we see resistivity changes, and this is verified by
direct current resistivity measurements conducted in 0.5 m depth intervals at multiple locations (not shown). The water
resistivity measurements were conducted using a 10 cm Wenner configuration. The measured resistivity of the water
column gradually varies from the top to the bottom of the lake, from ~27 to ~34 Ωm probably due to temperature
variations. For this reason, the water column was modeled with two resistivity layers with a priori constrained resistivity
values and a constrained water depth (depth to bottom of $2^{nd}$ water layer), but with a free interface between the two water
layers. Beneath the bottom of the lake (Profile AA' and BB' in Fig. 6), we observe sandy layers, underlain by a clay layer
interpreted to be Oligocene. The mean resistivity maps (Fig. 7c-f) at different depths reveal a large heterogeneity in the
geology below Lake Ravn. Along the shore of the lake, we observe sandy deposits, which most likely play an important
role in discharging groundwater to the lake.

### 4.2 Saltwater study


Horsens bay is a shallow fjord located in the western Baltic sea, Denmark, roughly 18 km long and 2-3 km wide. It has
poor ecological status, possibly due to submarine groundwater discharge causing excessive loading of nutrients (Hinsby
et al., 2012). Increased loading of nutrients has caused the Baltic sea as one of the most polluted seas in the world
(Pihlainen et al., 2020; Meier et al., 2019). To understand the vulnerability of the Horsens Fjord and coastal zone dynamics
an improved understanding of land-sea interactions including contaminant pathways in the subsurface, in relation to
nutrient and salinity variations, is needed.
The water depth within the survey area (Fig. 8) ranges from 2 m (minimum water depth for safe maneuverability with the
specific vessel) to 8 m in the central area. FloaTEM data were acquired in North-South striking lines across the bay





(Figure 8), with a line spacing of ~25 m and an operational speed of 12-14 km/h. The relatively small survey was
conducted in collaboration with the Geological Survey of Denmark and Greenland (GEUS). The purpose was to identify
and map fresh groundwater flow into the fjord, which may provide pathways for nitrate leaching from the surrounding
farmland into the bay.  The geology beneath the Horsens fjord includes quaternary meltwater sand and gravel constituting
as aquifer and quaternary clay tills and Miocene mica clay as aquitards (Jørgensen et al., 2010). A narrow channel connects
the fjord to deeper waters in the Baltic Sea. The central part of the fjord is dominated by muddy sediments due to the high
accumulation of organic material.  Till deposits are present in shallow coastal areas.
FloaTEM inversion results are presented in Fig. 9. The resistivity model in Horsens Bay (profile A in Figure 9) constitutes
a three-layer model where the top layer is the sea water followed by a conductive clay-rich infill sediments, likely an
extension of the Tørring/Horsens valley (Sandersen, 2016). The sequence is generally fining-upward, with significant
imprints of paleo-topography. Below the clay-rich layer, a third layer with elevated resistivity is present; interpreted as a
meltwater sand unit but saturated with sea water. The resistivity of this sand unit appears to be low (10-15 ohm-m)
compared to one would expect for fresh water saturated sand. This sandy unit is most likely leading the groundwater
discharge into the seabed at locations where the overlaying clay-till unit is sufficiently thin.
The mean resistivity maps (Fig. 9 b-e) show the spatial variability of the clay-till and sand rich sediments at four depth
intervals below the sea water label.  We see that the sediment close to the coast has a higher resistivity than what is
observed in the middle of the fjord. This might be a transition from a sandy sediment towards a more clay-rich
environment in the middle of the fjord. The knowledge of extension of these sand rich sediments from coast to the middle
of fjord, helps us to locate the probable regions where groundwater may discharge into fjord.   Additionally, we also
observe a small, northwest trending low resistivity structure indicates a paleo-channel, which has been confirmed by
shallow-seismic data (not shown).
**5. Discussion**
The resistivity of a surface water body can change over short distances, so inversions will often benefit from a spatially
varying resistivity constraint or reference.  The need for a priori water resistivity and depth is higher in the freshwater
cases than the saltwater case. The high conductivity in saltwater environments usually results in a well-resolved water
column, so a priori information is less important. While the current instrument is integrated with a depth sounder, it is not
difficult to fit it with a conductivity logger as well to supply relevant a priori values for the water column.  We note that
the choice in towing vessel is important as a larger vessel requires a longer towing distance.
In general, the data quality for FloaTEM is usually better than comparable land surveys as lakes and rivers are often far
from interfering infrastructure, which means that a FloaTEM survey normally results in full data coverage without gaps
from data culling.
FloaTEM data provide critical information regarding sub-lake or sub-sea geology. In the Lake Sunds example, an
interpretation based on land data only with lithological boundaries interpolated across the lake would be quite erroneous
by missing the unique features associated with the genesis of the lake. The FloaTEM system provides a means of capturing
these features which would be infeasible to identify with boreholes.



The depth of investigation is highly dependent on not only the resistivities of soils, but also of the conductivity of the
waters as the synthetic modeling study showed, where even a small conductivity change in the saltwater can reduce the
DOI significantly. This stresses that a priori information about water salinity values is critical in selecting between the
FW-FloaTEM and SW-FloaTEM configurations and designing the particular survey design.
The high signal level in conductive saltwater environments often results in very low noise, also at the latest recorded time
gate at ~2 ms. In these cases, increasing the recording time and reducing the repetition rate should increase the DOI by
adding more late-time data. However, a lower repetition rate may also lead to higher motion induced noise in the receiver
coil, which can become the dominating noise for the late time gates.
The results showed here all focused on delineation of hydrological permeable (sands) and impermeable (clays) lithologies
in the context of improving large-scale hydrological understanding and prediction strength. Though, from the given
examples it should be clear that the application range of FloaTEM spans much more. A few examples include foundation
investigations for offshore wind farms; raw material exploration beneath lakes and rivers; and geotechnical pre-
investigations for cabling routes below water bodies.

## 6. Conclusions

We have developed a new towed, easily configurable floating TEM instrument, FloaTEM, and successfully applied the
system to both freshwater and saltwater studies to investigate geology and hydrology beneath lakes and shallow seawater.
The FloaTEM system is modular, so longer beams can be used to increase the transmitter moment and likewise more
transmitter turns can be added, both increasing the depth of investigation. Supported by synthetic analysis, we
reconfigured a freshwater FloaTEM system to a saltwater FloaTEM system, primarily by increasing the transmitter
moment and decreasing the noise in the receiver coil enabling us to perform FloaTEM surveys not only on both shallow
and deep lakes, but also on shallow saltwater up to 8 meters deep.
The conductance of the water, water depth multiplied with water conductivity, is the limiting factor when surveying on
saline water. Based on the presented analysis the water column should be below ~25 Siemens for the system to penetrate
the water column and map sub-water layers. For freshwater lakes and rivers, depths of investigation of 80 meters or more
are possible, while in saltwater cases we can achieve depths of investigation of 10-25 meters strongly depending on water
depth and conductivity.
With the FloaTEM system, we can map geological layers beneath the water bodies, normally not accessible for mapping
with ground based geophysical methods, thereby allowing for detailed hydrological modelling in these often-important
areas as well. Through 2 freshwater cases and one saltwater case we show the system's ability to image the heterogeneous
geology beneath water bodies. In the freshwater cases the FloaTEM datasets revealed geological information that would
have been impossible to deduce from land-based-only information and in the saltwater case the data delivered clear images
on the clay-sand distribution beneath the seafloor.

## 7. Author contribution

PM design and develop methodology, instrumentation, data processing and inversion, wrote the first draft of manuscript.
FC carried out data collection, data analyses and contributed to original manuscript. JP and MK contributed to first draft
of the manuscript and interpretations and feedback on inversion results. RF provided data interpretations, feedback and



contributed to the writing of original manuscript. NF carried out synthetic data analysis and field data inversion of Ravnsø
lake. AV and EA conceptualized the methodology, contributed to writing original manuscript and provided feedback.

**8. Acknowledgements**

We thank TOPSOIL, an Interreg project supported by the North Sea Programme of the European Regional Development
Fund of the European Union, and the development has been funded by Innovation Fund Denmark, project rOpen (Open
landscape nitrate retention mapping) and MapField (Field-scale mapping for targeted N-regulation), WATEC (Aarhus
university Centre for water technology) and internal HGG (Hydrogeophysics group at Arhus university) funding. Partial
support for data collection and interpretation of results were provided by GEUS (Geological survey of Demark and
Greenland).

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

**List of Figures**

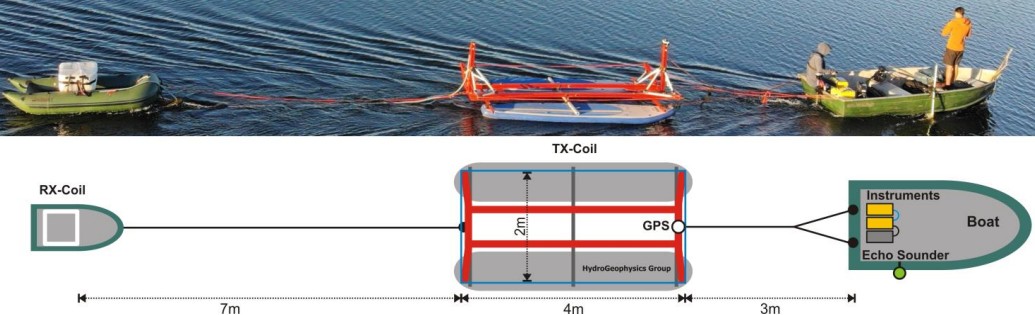

**Figure 1: Picture and schematic of the freshwater FloaTEM configuration, with boat, transmitter coil (TX-coil), and receiver**
**coil (RX-coil). In contrast, the saltwater configuration uses a 4m x 4m transmitter coil.**

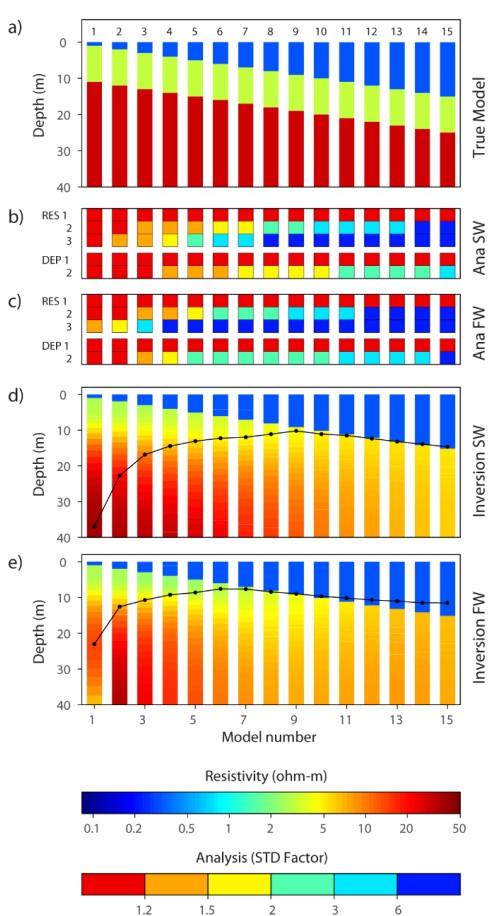

**Figure 2. a) True model. Number on top of each model bar states the water depth (thickness of first layer). b-c) Model parameter analyses of the true models, stated as a standard deviation factor, for the SW- and FW-FloaTEM systems. d-e) Inversion results for SW- and FW-FloaTEM systems. The black line shows the DOI.**

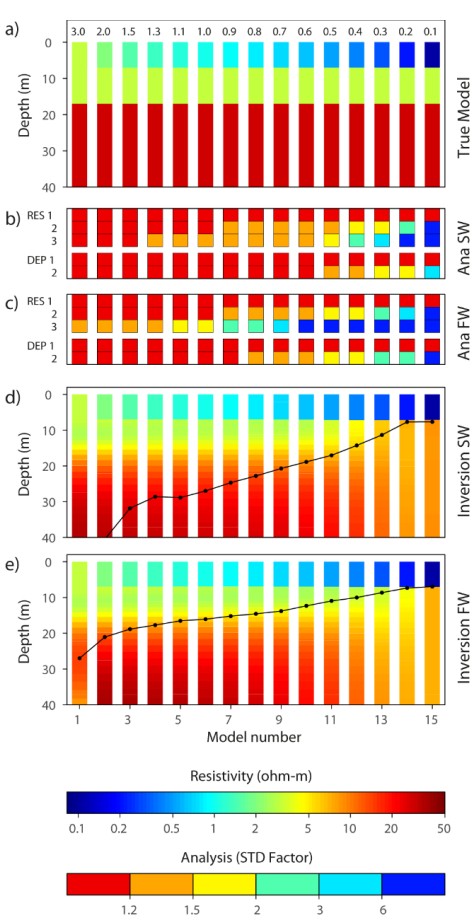

441

Figure 3. Model sweep 2. a) True model. Number on top of each model bar states the resistivity of the water (resistivity of first layer). b-c) Model parameter analysis of the true model, stated as standard deviation factor, for the SW- and FW-FloaTEM systems. d-e) Inversion results for SW- and FW-FloaTEM systems. The black line shows the DOI.





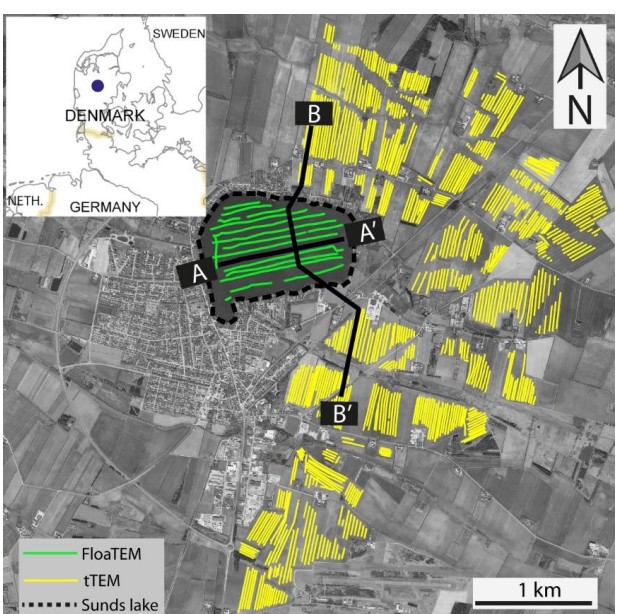

445

Figure 4 Sunds FloaTEM and tTEM survey region with FloaTEM lines marked in green and tTEM in yellow. AA' and BB'
are the profiles that are presented in Figure 5.

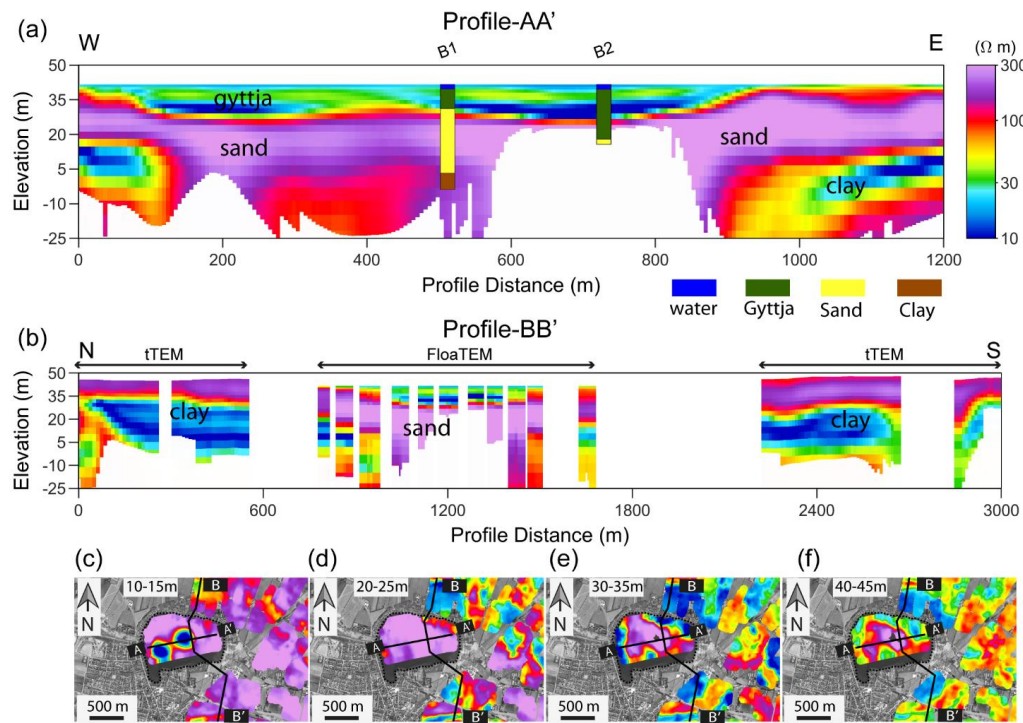

448

**Figure 5: Results from Sunds joint tTEM and FloaTEM survey. Location of profile AA' and BB' is marked in** Error! Reference
source not found.**; note that while the elevation axis is identical, the profiles have different lengths and thereby different vertical
exaggeration. Profile-AA' includes lithological interpretations from available boreholes near the survey line. Note that the
water column is included in the figure, but only 2 meters thick, (c) - (f) show mean-resistivity maps at various depth intervals
with profile -AA' and BB' indicated as solid black line. Lake Sunds is marked with dotted black line.**



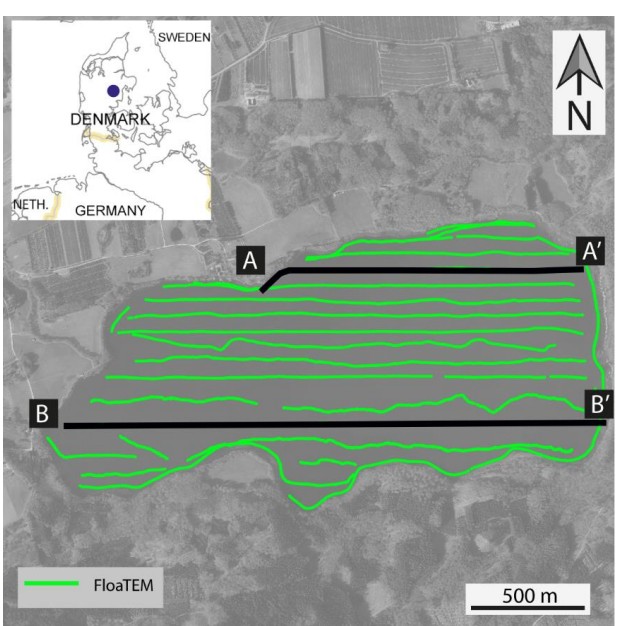


**Figure 6: Survey region for the Lake Ravn FloaTEM survey. Locations of the profiles in Figure 7 are highlighted as solid black lines**

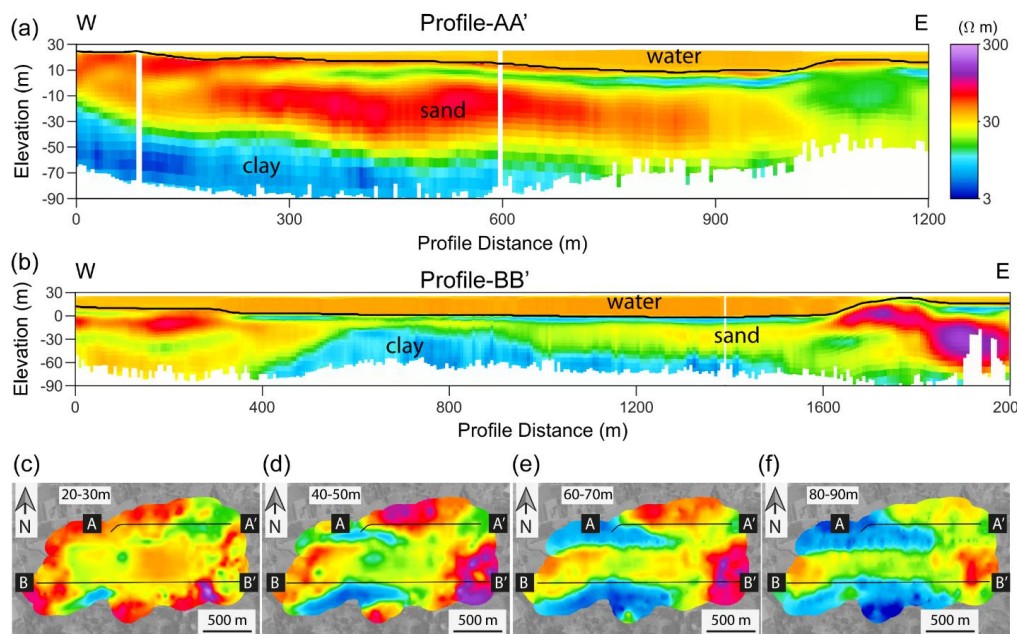


**Figure 7: Results from Lake Ravn FloaTEM survey. Location of the resistivity sections AA' and BB' are marked in** Error!
Reference source not found.**. The black line in the sections marks the lake bottom while the white faded colors indicate the DOI.**
**(c) – (f) show mean-resistivity maps at 4 depth intervals below surface together with location of profile AA' and BB'.**





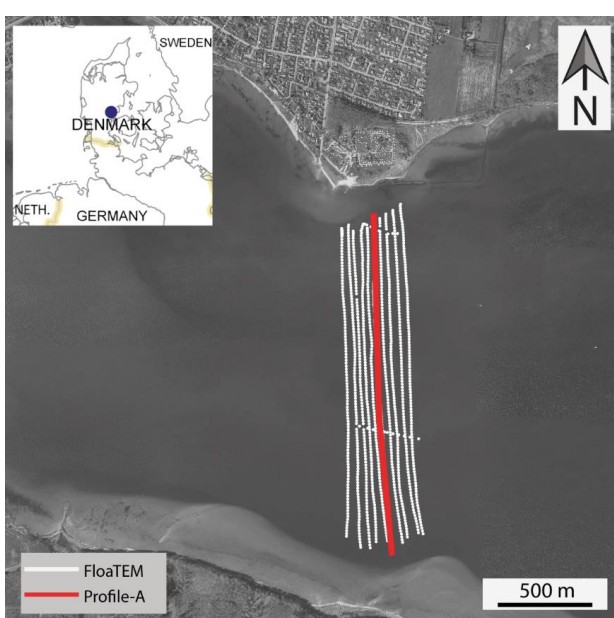


**Figure 8: Horsens Bay with FloaTEM survey lines. The red highlighted profile marks the location of the resistivity section showed in figure. 9**





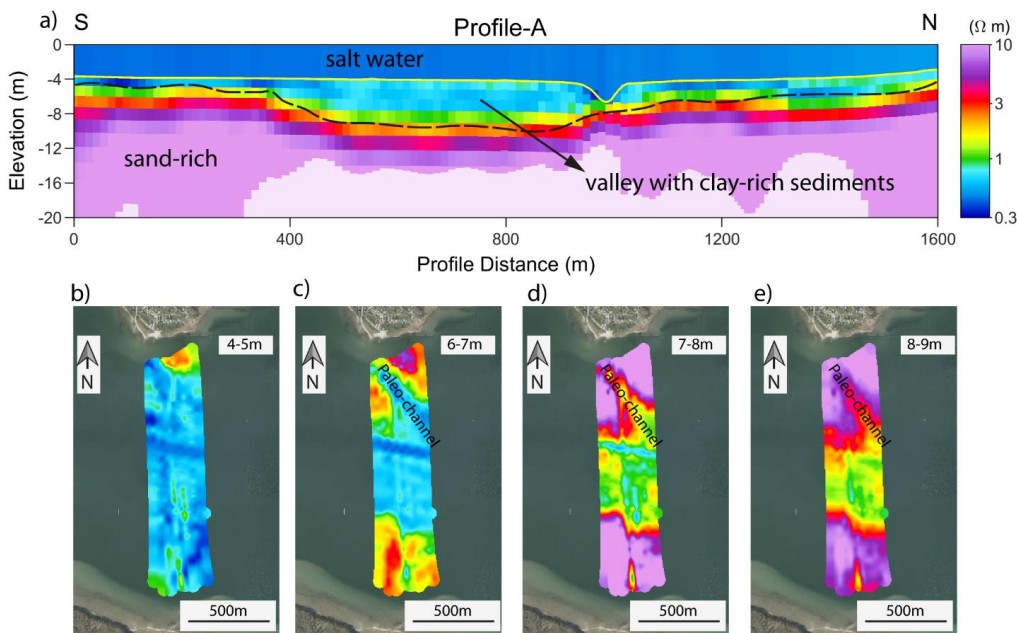

**Figure 9: Resistivity mapping results from Horsens Bay. (A) Resistivity section (location marked in Figure 8) with the seafloor marked with the yellow line. (b-e) Mean resistivity maps at different depths.**

**List of Tables**

| FloaTEM system | FW-FloaTEM | | SW-FloaTEM |
|---|---|---|---|
| | Low moment | High moment | High moment |
| Transmitter area | 8 m² | | 16 m² |
| Number of turns | 1 | | 4 |
| TX peak current | ~3 A | ~30 A | ~25 A |
| TX peak moment | ~24 Am² | ~240 Am² | 1600 Am² |
| Repetition frequency @ 50 Hz power line frequency | 2110 Hz | 630 Hz | 220 Hz |
| Duty cycle | 42% | 30% | 22% |
| Tx on-time | 200 µs | 450µs | 1000 µs |
| Turn-off time | 2.6 µs | 4.5 µs | 14.10 µs |
| Gate time interval (from beginning of turn-off) | 4-33 µs | 10-900 µs | 20-2800 µs |
| *RX coil area* | 20 m2 | 20 m2 | 40 m |
| *RX coil bandwidth* | 420 kHz | 420 kHz | 140 k Hz |
| Number of gates | 15 | 23 | 26 |

**Table 1: System parameters for the freshwater and saltwater FloaTEM systems.**





| Survey area | Max. water depth | System | Line spacing nominal | Water depth prior constraint | Water resistivity, prior constraint |
|---|---|---|---|---|---|
| **Lake Sunds** | 4.5 m | FW-FloaTEM | 50 m | 1.03 | 15 Ωm, None |
| **Lake Ravn** | 34 m | FW-FloaTEM | 60 m | 1.05 | *28 Ωm, 1.1 <br> *34 Ωm, 1.05 |
| **Horsens fjord** | 8 m | SW-FloaTEM | 35 m | 1.05 | 0.3 Ωm, None |


**Table 2: Survey configurations and conditions of the three case areas. The * indicates that the water column was modeled with two resistivity layers.**


**Appendix-A**
**Data processing and inversion**
In this section, we give an overview of the data processing and inversion scheme used for FloaTEM data. In each of the
case studies, FloaTEM data were processed with the Aarhus Workbench software from Aarhus GeoSoftware
(www.arhusgeosoftware.dk). The standard FloaTEM processing flow follows Auken et al. (2009). Raw db/dt data are
first processed to remove coherent coupling interference due to nearby infrastructures and then stacked to produce
soundings with approximately 10 m spacing. In the presented cases, a short smoothing filter was applied on the recorded
water depth data, but this step depends on the quality of the depth sounder data at hand. A preliminary inversion is then
performed to evaluate and adjust the first-step processing of raw db/dt data.
The final inversions of the FloaTEM data were carried out using a spatially constrained inversion formulation, SCI
(Viezzoli et al., 2009) using a 30 layer smooth model with layer thicknesses of layers 2-30 increasing logarithmically
down to 120 m. The thickness of layer 1 is set to the water depth with a fairly tight prior constraint. No vertical resistivity
constraints are applied from the water layer (layer 1) to the sub-layers (layers 2-30), hereby allowing a shape boundary at
the lake-/ seabed in the inversion results. The water depth prior information can be taken from the echo-sounder data or
from an external bathymetry grid. Additionally, prior constraints can be added to the resistivity of the water layer if
separate measurement of the water conductivity are present. In some cases, it is insufficient to model the water column
as one homogeneous layer, e.g. probably due to a halocline or thermocline. In these cases, more layers are introduced to
represent the water column in the inversion setup and the prior water depth is assigned to the depth to the bottom of the
last water layer.