# Peer review of "Technical note: Efficient imaging of hydrological units below lakes and fjords with a floating, transient electromagnetic system (FloaTEM)"

_Hydrology and Earth System Sciences, 2021_

## Author Response (AR1)

**Response to Reviewer-1**

Dear Maurya et al.,

I commend you on this great paper about the FloatTEM system. As a practitioner of EM geophysics in the geotechnical engineering sector, these case studies and synthetic modelling studies are very helpful illustrations about the capabilities of your new system. I particularly like the very clear and straightforward formulation of the result of that synthetic modelling study: 25 siemens is the limiting conductance on the SW-FloatTEM.

I could only find a few minor points where I could think of improvements to your paper, mostly in the Discussion section of your paper and about the limitations of your system beyond DOI.

*Dear Craig,*

*Thank you for taking time and giving feedback on the paper. Please see below the response of your comments/suggestions.*

First, your synthetic modelling studies show that the freshwater system lacks the same DOI as your saltwater system. Given that, why not just deploy the saltwater system in all cases? Are there disadvantages in using the SW-FloatTEM? Do you lose lateral resolution? Are there operational challenges in getting equally good data?

*Response: We cannot deploy the SW-FloaTEM system in all cases. Because SW-FloaTEM has a longer turnoff since it has a bigger loop and more turns, it therefore doesn't provide unbiased early time signal usually before 10 us, which in freshwater case results in loss of resolution in first 0-20 m.*

Second, I have one concern that you rely heavily on a priori knowledge of water resistivity and depth when making models. As a practitioner, I can imagine bathymetry or conductivity meters failing. Does your synthetic study offer any insights into how this might affect results? Are there certain environments, like rivers entering the sea, where spatial variation in electrical resistivity is very high and extra considerations are needed?

*Response: I agree, in this paper we have included priori knowledge of water resistivity and depth assuming that all functionality of the system is operational. However, depending on the environments, local geological conditions, the system would still perform well and provide you satisfactory and useful inversion models in the absence of water resistivity and depth information. This two informations are additional but not mandatory for inversion and provide better resolution. In case of lateral resistivity variation, you can let the resistivity of the water column be free in inversion and only constrain the water depth. This will allow the inversion to adopt any lateral resistivity variations .*

Similarly, on Line 132, how did you come up with the 10 % and 30 % uncertainty for water depth and resistivity, respectively? Do you have a source to back up these choices? Are these the realistic uncertainties in real-world instruments

*Response:*

*We normally know the water depth from the echo-sounder or similar and the water conductivity by measurements with the conductivity probe, and we use this info as a prior in the inversion of real data. Therefore, we also add priors on water depth and resistivity for the synthetic studies. The 10% and 30% (0.1, 0.3) prior constraints are more conservative estimates to be on the safe side. A single water depth/conductive measurement are less uncertain than 10/30%, but there are point measurements while TEM has a footprint. However, for field cases as in this paper, we used 5 -10%. In synthetic cases we tried to be conservative , so that we don't over-estimate the model resolution.*

Lastly, you haven't commented on the thin conductor between lakewater and sand in your Lake Ravn case study (Figure 7). Do you believe this is sand or something else (like lakebed, organic sediments)?

*Response:  Yes, this thin layer we believe is fine sediments like clay/silt.  Thanks for pointing out, we will include this in the revision.  We inserted this interpretation the line 240-241:as below*

*"Below the bottom of the lake, we observe a thin conductive layer which is interpreted as fine sediments deposits such as clay or silt."*

My strength is more on the interpretation of geophysical models to EM theory, so I would have to rely on other reviewers to give more detailed comments about the setup of your system. But, to my knowledge, I don't see any obvious shortcomings.

Otherwise, I only have minor comments:

- Line 81: I would revise to "in the following subsections"

- I am rather picky about hyphenation of multi-word adjectives. Here's a helpful source on the matter:
https://owl.purdue.edu/owl/general_writing/punctuation/hyphen_use.html . Some instances in your paper where I would revise:

  -- Line 76: real-time

-- Line 202: 70-year-old (note singular use of "year")

- Line 80: "freshwater" should be a single word

- Capitalize proper names of bodies of water. For example, Line 261: "Horsens Valley"

- There was an issue with the cross-reference pointing to other figures with maps at Lines 449-450 and 458-459

- Table 1: Incorrect unit for RX coil area for SW-FloatTEM

- Line 229: I was going to suggest a less whimsical-sounding word than "wigglier" to use, but I can't think of a more formal word to use in its place. So I suppose this fun word can stay.

 *Response:   Thanks for these minor suggestions, we will modify them in the revised manuscripts.*

Thank you again and I look forward to seeing your final version being officially published.

**Response to Reviewer-2**

Dear all,

This paper introduces a new towed floating TEM instrument through model studies and several case studies. The structure is very clear of this paper and straightforward. I think the authors have taken good studies to illustrate the capabilities of the new system.

I read the comments from the first referee Craig and the authors' responses. Some of my questions have been answered. Thus, here I only list the comments which I still hold.

*Response: Dear Shuangmin Duan, thank you for taking time and review the paper. Please see below the response to your comments.*

Except for the application results of the instrument, I also expect more figures about the technical parts. Beyond table 1, I would prefer to see a figure of the waveform of the transmitted current and in Appendix-A I have expected to see your processing result of the transients, the error bars and the data fitting of your inversion, which can also provide a lot of information about how your instrument performs.

*Response: in Appendix A, we have added a figure of the waveforms and a figure showing processed data with error bars and data fitting of the inversion.*

When designing the instrument, you considered the water resistivity, depth, the TX current moment and so on to get a higher DOI. But what is the influence of the RX-TX offset on DOI. Why is a 9 m offset? How do you estimate DOI for these inversion examples?

*Response: The Rx-TX offset is chosen as small as possible to make the system compact. The RX is placed 9m offset from the TX to minimize the interference from TX to the RX-coil. We have not investigated RX-TX offset influence on the DOI, but the primary control of the DOI is late time gates, water resistivity, depth, the TX current moment. DOI is calculated following the Christiansen et.al., (2012) method; we included the reference in the revised manuscripts (Line 129)*

In the instruction part, the author should also include the surface towed and deep dragged EM setups for hydrologic applications, such as groundwater explorations in

shallow sea areas presented by Micallef et al., (2020) and Gustafson et al., (2019). Maybe you could find more.

Response: the above refences has been included in revised manuscripts. (Line 54-55)

Line 107 and 109: spelling 40m2->40m2

Response: will be changed in revised manuscripts

Line 117: "The model resolution study was also used in the design of the **SW-FloaTEM** system", I guess here you want to say, "The model resolution study was also used in the design of the **FW-FloaTEM** system"?

Response: We want to emphasize that the design of the SW-Floatem system was based on the model resolution study. For Comparison purpose we also included the model resolution study of FW-FloaTEM system. We clarify this in the revised manuscripts ad below: Line (119-120)

*"Conclusion derived from the model resolution study led to the design of the SW-FloaTEM system. We also present the analyses of the FW-FloaTEM system to compare against the SW-FloaTEM system."*

Line 133-135: I am confused with this expression "For the inversion, no lateral constraints were applied. However, for the model parameter analysis lateral constraints were assumed between 5 similar neighboring models (based on the true model) to simulate the improved resolution capabilities from information sharing when working with field data." Since this is a 1D inversion, how do you use lateral constraints? And what do you mean here on "model parameter analysis"?

*Response: we follow the spatially constrained inversion algorithm by Viezzoli et al., (2009). All 1D inversion models are inverted by minimizing one objective function which includes lateral constraints between neighboring model parameters. Model parameter analyses shows how well a model parameter (resistivity or depth) likely to be resolved by inversion. See Auken et, al. (2015) for the detailed explanation.*

Figure 5: The y-axis is confusing. What does positive and negative elevation mean? Where is the water depth? Below profile A-A', you make some legend as water, Gyttja, Sand and Clay. This makes me confused at the beginning since in the figure you use purple as the sand layer but in the legend you use yellow. But later I realized the legend

only serves for the drilling B1 and B2. Maybe you could find a better way to display them.

*Response: Elevations above sea level are positive and elevations below sea level are negative. The revised manuscripts have new figures where we specifically mention the Borehole legends to avoid the confusion.*

Figure 5: Do you use interpolation to display profile A-A', since it is laterally smooth. If yes, what kind of interpolation do you use and what side effect this will bring? Why between 600-800 m, the DOI is so shallow, limited only to the upper 20 m?

Response: There is no interpolation along the profile A-A'. The reason why it looks smooth because it is drawn along a survey line, following approximately ~10 m spaced models, therefore, models are placed very close, giving a smooth nature.

References:

Auken, E., Christiansen, A. V., Westergaard, J. A., Kirkegaard, C., Foged, N., and Viezzoli, A.: An integrated processing scheme for high-resolution airborne electromagnetic surveys, the SkyTEM system, Exploration Geophysics, 40, 184-192, 2009.

Viezzoli, A., Auken, E., and Munday, T.: Spatially constrained inversion for quasi 3D modelling of airborne electromagnetic data - an application for environmental assessment in the Lower Murray Region of South Australia, Exploration Geophysics, 40, 173-183, 2009.

Christiansen A. V, Auken E. A global measure for depth of investigation. Geophysics. 2012 Jul 1;77(4):WB171-7.